# The Thousand Faces of Malignant Melanoma: A Systematic Review of the Primary Malignant Melanoma of the Esophagus

**DOI:** 10.3390/cancers14153725

**Published:** 2022-07-30

**Authors:** Gerardo Cazzato, Eliano Cascardi, Anna Colagrande, Teresa Lettini, Leonardo Resta, Cinzia Bizzoca, Francesca Arezzo, Vera Loizzi, Miriam Dellino, Gennaro Cormio, Nadia Casatta, Carmelo Lupo, Antonio Scillimati, Salvatore Scacco, Paola Parente, Lucia Lospalluti, Giuseppe Ingravallo

**Affiliations:** 1Section of Pathology, Department of Emergency and Organ Transplantation (DETO), University of Bari “Aldo Moro”, 70124 Bari, Italy; anna.colagrande@gmail.com (A.C.); teresa.lettini@uniba.it (T.L.); leonardo.resta@uniba.it (L.R.); giuseppe.ingravallo@uniba.it (G.I.); 2Department of Medical Sciences, University of Turin, 10124 Turin, Italy; eliano20@hotmail.it; 3Pathology Unit, FPO-IRCCS Candiolo Cancer Institute, Str. Provinciale 142 lm 3.95, 10060 Candiolo, Italy; 4Section of General Surgery, Azienda Ospedaliero-Universitaria Policlinico di Bari, 70124 Bari, Italy; cinzia.bizzoca@policlinico.ba.it; 5Section of Gynecology and Obstetrics, Department of Biomedical Sciences and Human Oncology (DIMO), University of Bari “Aldo Moro”, 70124 Bari, Italy; francesca.arezzo@uniba.it (F.A.); vera.loizzi@uniba.it (V.L.); miriamdellino@hotmail.it (M.D.); 6Oncology Unit IRCSS Istituto Tumori “Giovanni Paolo II”, Department of Interdisciplinary Medicine (DIM), University of Bari “Aldo Moro”, 70124 Bari, Italy; gennaro.cormio@uniba.it; 7Innovation Department, Diapath S.p.A., Via Savoldini n.71, 24057 Martinengo, Italy; nadia.casatta@diapath.com (N.C.); carmelo.lupo@diapath.com (C.L.); 8Department of Pharmacy-Pharmaceutical Sciences, University of Bari “Aldo Moro”, 70124 Bari, Italy; antonio.scillimati@uniba.it; 9Department of Basic Medical Sciences and Neurosciences, University of Bari “Aldo Moro”, 70124 Bari, Italy; salvatore.scacco@uniba.it; 10Unit of Pathology, Fondazione IRCCS Ospedale Casa Sollievo della Sofferenza, 71013 San Giovanni Rotondo, Italy; paolaparente77@gmail.com; 11Section of Dermatology and Venereology, Department of Biomedical Sciences and Human Oncology (DIMO), University of Bari “Aldo Moro”, 70124 Bari, Italy; l.lospalluti@gmail.com

**Keywords:** malignant melanoma, esophagus, primary, neoplasia, skin, mucosal melanoma, genetics

## Abstract

**Simple Summary:**

Primary malignant melanoma of the esophagus (PMME) represents a fairly described and represented entity in the literature, although only in the middle of the last century was it recognized as actually existing. In this systematic review we address the history of PMME starting from 1906, the year in which Bauer made the first description of this entity, passing through his first demonstration on an autopsy series by De Pava et al. and we come to the most recent notions of molecular biology and immunotherapy.

**Abstract:**

Primary Malignant Melanoma of the Esophagus (PMME) is an extremely rare cancer of the esophagus, accounting for 0.1–0.8% of all oro-esophageal cancers and <0.05% of all melanoma subtypes, with an estimated incidence of 0.0036 cases per million/year. We conduct a careful analysis of the literature starting from 1906 to the beginning of 2022, searching the PubMed, Science.gov, Scopus and Web of Science (WoS) databases. A total of 457 records were initially identified in the literature search, of which 17 were duplicates. After screening for eligibility and inclusion criteria, 303 publications were ultimately included, related to 347 patients with PMME. PMME represents a very rare entity whose very existence has been the subject of debate for a long time. Over time, an increasing number of cases have been reported in the literature, leading to an increase in knowledge and laying the foundations for a discussion on the treatment of this pathology, which still remains largely represented by surgery. In recent times, the possibility of discovering greater mutations in gene hotspots has made it possible to develop new therapeutic strategies of which nivolumab is an example. Future studies with large case series, with clinicopathological and molecular data, will be necessary to improve the outcome of patients with PMME.

## 1. Introduction

Primary Malignant Melanoma of the Esophagus (PMME) is an extremely rare cancer of the esophagus, described for the first time by Bauer in 1906 [1] accounting for 0.1–0.8% of all oro-esophageal cancers and <0.05% of all melanoma subtypes [2,3] with an estimated incidence of 0.0036 cases per million/year [4]. Between the report of the first case and 2021, 347 cases of PMME were reported. PMME constitutes 5.6% of all gastrointestinal (GI) tract melanomas. Although PMME is a rare subtype of esophageal cancer, it has been relatively well reviewed. PMME is a highly aggressive tumor that is associated with a poor prognosis, with a 5-year overall survival (OS) rate of <5% [5,6]. The debate on its genesis starting from the esophagus has been quite heated since it was not believed at the time of the first descriptions that there could be melanocyte cells at the esophageal mucosal level [4,5,6,7,8,9,10,11,12,13,14,15,16,17,18,19], and only with the first autoptic series did it begin to prove in a clear way that melanoma could also originate from this part of the body [20]. From a topographical point of view, 90% of PMME cases involve the middle/distal third of the esophagus, which is usually solitary, but in about 13% of cases it presents with multiple lesions. Macroscopically, these are intra-luminal polypoid lesions, capable of obstructing the esophageal lumen in a variable degree percentage, with symptoms related to the degree of stenosis. In more than 85% of cases, PMME presents as a hyperpigmented lesion, and in the remaining 15% of cases it presents as amelanotic. One of the most important differential diagnoses is with esophageal melanosis (or melanocytosis), which would represent a pre-neoplastic lesion for some authors [20,21,22,23,24,25,26].

In this paper, we conduct a careful analysis of the literature starting from 1906, the year in which Bauer described a case of PMME for the first time, and we arrive at the beginning of 2022, focusing on the new therapeutic strategies of the latest generation, such as immunotherapy, trying to designate future perspectives on the treatment of this entity.

## 2. Materials and Methods

A systematic review was performed following the Preferred Reporting Items for Systematic Reviews and Meta-Analyses (PRISMA) guidelines. Indeed, a search of the PubMed, Science.gov (accessed on 17 July 2022), Scopus and Web of Science (WoS) databases was performed for the period 1906–2022 using the following terms: Primary or Primitive Malignant Melanoma or Melanoma in combination with each of the following: Esophagus, Oesophagus, Mucosal. The last search was run on 8 May 2022. Only manuscripts reporting certain cases of primitivity of melanoma in the esophagus were included. Articles with uncertainty about primitiveness or whose metastasis to the esophagus could not be ruled out were not considered. Review articles with/without meta-analyses, observational studies and case reports were all included, without language distinction. Other potentially relevant articles were identified by manually checking the references of the included literature.

An independent extraction of articles was performed by two investigators (G.C. and K.M.) according to the inclusion criteria. Disagreement was resolved by discussion between the two review authors.

## 3. Results

A total of 457 records were initially identified in the literature search, of which 17 were duplicates. After screening for eligibility and inclusion criteria, 303 publications were ultimately included (Figure 1). The majority of publications were case reports (*n* = 273), followed by review (*n* = 13), case series (*n* = 5), observational prospective studies (*n* = 4), letters or editorial (*n* = 6) and comparative studies (*n* = 2). A total of 347 patients’ PMMEs were recorded.

The most common symptoms were dysphagia (75%), retrosternal pain (69%), difficulty swallowing solids and liquids (51%) and, in the most advanced cases, weight loss and digestive difficulties, along with more rare melena and hypovolemic shock. The median survival, according to the various authors, was estimated from 2 to 5% at 5 years, with a median survival of 10 months. A total of 90% of cases involved the middle and distal thirds of the esophagus; in 88% of cases it was a solitary lesion, but in the remaining 12% of cases, multiple lesions were reported. Macroscopically, these lesions were usually intra-luminal polypoids with variable obstruction of the esophageal lumen. In 85% of cases, they turned out to be pigmented, with about 15% being amelanotic. From the works included in these reviews, it appears that the main differential diagnoses are represented by esophageal melanocytosis, both from a macroscopic/endoscopic point of view and a histopathological point of view. In most of the reported cases, PMME involved the mucosal and submucosal plane, with frequent radial growth and frequent invasion of the lympho-vascular space. In about half of the cases analyzed (50%), the disease was already metastatic to other organs, represented (in decreasing order of frequency) by: liver, mediastinum, lung and brain.

## 4. Discussion

Although a rare entity and considered anecdotal for a long time, PMME has been studied and described over the decades, with different levels of description ranging from single case reports of “pigmented” lesions in correspondence with the esophageal mucosa, up to actual studies of clinicopathological and molecular characterization of this primary esophageal neoplasm [4,5,6]. In 1952, Garfinkle J. M. et al. reported the first case with histological correlation described in the literature. In fact, the previously reported cases had only been illustrated from a clinical point of view, with no real demonstration of malignant melanocytic neoplasm. In this paper, the authors described a pigmented lesion of the esophageal mucosa of a 55-year-old woman, on which biopsies suggesting a melanocytic nevus were taken. Only subsequently did the morphological finding indicate a malignant melanoma [7]. In the following years, other authors [8,9,10,12,13,14,15,16,17,18,19] reported their experience with suspected or presumed cases of PMME, even if the number of cases that could actually be considered primitive of the esophagus were even more limited than those published until then, as the Allen–Spitz diagnostic criteria [11] restricted the effective number of PMMEs to: (1) typical histologic pattern of melanoma with the presence of melanin granules within the tumor cells; (2) origin in an area of junctional change within the squamous epithelium; (3) junctional activity with melanotic cells in the adjacent epithelium). In 1957, with the paper by Keeley et al. [12], this entity began to be defined as malignant melanoma of the esophagus and not with terms such as “melanocarcinoma” or “melanosarcoma” or “nevocarcinoma” or “melanoblastoma” of the esophagus [13,14,15,16,17,18,19,20,21,22,23,24,26,27,28,29].

In 1963, de la Pava et al. [20] showed the presence of melanocytes in the esophageal epithelium in four out of 100 normal esophagi from autopsied materials and only from this moment was the actual existence of PMME officially recognized.

In 1970, Piccone et al. published an interesting case report concerning a patient, a 70-year-old man, with a primary melanosis of the esophagus and in whom a PMME had developed. The authors reported that this was the fourth case in the literature of a patient with melanosis of the entire esophagus and supported their hypothesis of an origin in the context of esophageal melanosis, studying the different pigmented lesions affecting the subject [25]. In 1972, Frable et al. reported one of the first electron microscopy studies performed on PMME tissue. Cancer cells contained membrane-bound dense material consistent with melanin and a rare pre-melanosome. The latter structure was not typical, lacking the usual crystalline pattern or cross-linking. This last structure was interpreted by the authors as a stage in the development of melanin, closer to the mature product [30]. These types of studies have been fundamental to understanding, over time, if and how it was acceptable to assert a primitive origin of a malignant melanoma starting from the esophagus.

Over the years, PMEE reports have multiplied [26,27,28,29,30,31,32,33,34,35,36,37,38,39,40,41,42,43,44,45,46,47,48,49,50,51,52,53,54,55,56,57,58,59,60,61,62,63,64,65,66,67,68,69,70,71,72,73,74,75,76,77,78,79,80,81,82], and, in 1983, Aldovini et al. [56] reported their experience with a 54-year-old male patient with a PMME on which a brushing cytology had been performed, which revealed atypical cells with melanin granules that tested positive with the Fontana–Masson method. In this paper, we can glimpse the growing certainty of the real existence of PMEE, considering that around the brownish neoplastic mass, the presence of irregular pigmentation (corresponding to junctional snow lesion) was observed. In 1990, Taniyama et al. [83] described a very rare case of amelanotic PMEE in a 76-year-old patient with a polypoid lesion with an irregular, whitish surface, located in the intrathoracic middle third of the esophagus. The authors discussed the possible difficulties in recognizing this type of melanic lesion without pigment and stressed the need to: (1) carefully search for possible lesion components at the dermal-epidermal junction and (2) perform appropriate immunohistochemical investigations to demonstrate the melanocytic nature of the lesion component. Regarding the epidemiology of PMEE, in a 1990 paper by Aagaard et al. [84], a single center experience of 25 years related to primary non-epithelial malignant tumors of the esophagus was reported. The authors reported on a total of 708 tumors of the thoracic esophagus, and only 4 cases (0.56%) of PMME. In other papers [85,86,87,88,89,90,91,92,93,94,95,96,97,98,99,100], Chello et al. [101], in 1993, reported the case of a metastatic PMME to the walls of the left atrium of the heart in a 75-year-old patient with massive pericardial effusion. The authors of the article stressed the importance of excluding metastatic disease in subjects with melanoma, meeting the now acclaimed Allen–Spitz criteria for a correct diagnosis of PMME, and moreover, stressed the importance of keeping in mind the potential of malignant melanoma to be able to metastasize, sometimes even at the level of the heart. In pursuing this aim, they mentioned Waller et al., who suggested that in a patient with a known history of MM, an acute pericarditis, or sudden pericardial effusion, a second or third degree atrioventricular block, or cardiac tamponade, or also congestive heart failure, healthcare professionals should always suggest excluding the possibility of metastases of melanoma.

Over the next nine years (1993 to 2002), another 50 papers [102,103,104,105,106,107,108,109,110,111,112,113,114,115,116,117,118,119,120,121,122,123,124,125,126,127,128,129,130,131,132,133,134,135,136,137,138,139,140,141,142,143,144,145,146,147,148,149,150,151,152] reported 62 cases of PMME, with some peculiarities represented, among others, by another case of amelanotic PMME [106] by Loftus et al., and then a case of PMME with a massive hememesis [108] in which it was emphasized that, although rarely, PMME can present itself primarily with a hyperacute symptomatology and that puts the life of the patient in danger. Interestingly, Yano et al. reported the association of a PMME with an adenocarcinoma of the lung for the first time [128]. Equally interesting was the paper by Lam et al., which reported a case of PMME with loss of immunoexpression of p53, going to try to analyze (albeit in an embryonic way) the mechanisms of gene deregulation that are also found during this neoplasm [135].

In 2002, Brown et al. [153] reported the challenging case of a 76-year-old woman who had had dysphagia and retrosternal pain for about three months. The first diagnosis, carried out on biopsy, had suggested the possibility of a PMME, even if the lesion was not easy to classify nosographically. After the following esophagectomy, the sample was studied in detail, and it was realized that more than a PMME, it was a melanotic schwannoma, much rarer than PMME (about 11 cases described up to the time of the article in question) but reasonably related to malignant melanoma from the point of view of differentiation.

After the paper by Holck, Lin and Benferhat et al. [154,155,156], in a paper by Lohmann C. et al. [157] in 2003, a case series of 10 patients (mean age of 64 years) with PMME was presented. Most of the lesions reported were located in the mid-distal esophagus and were large, with a mean tumor size of 6.2 cm at diagnosis and a mean invasion depth of 1.86 cm); furthermore, two melanomas were associated with a large component in situ and about half of the tumors were amelanotic. Histological features were broad, including cases that mimicked lymphoma, poorly differentiated adenocarcinoma or sarcoma. In this study, 7/10 patients had metastatic disease at the time of presentation, and 9/10 patients underwent tumor resection with negative surgical margins. The authors reported a poor survival rate, with a median of 19.8 months. On the other hand, one patient with a tumor limited to the submucosa was still alive 108 months after esophagectomy. A total of 6/10 patients with PMMEs were tested by immunohistochemistry and were positive for all melanocyte differentiation markers; but also were positive for CT antigens, with MA454 being the most commonly found, suggesting that CT antigens could have been a promising immunotherapeutic target for esophageal melanomas.

PMME cases continued to increase in number, reaching about 260 in 2006 [154,155,156,157,158,159,160,161,162,163,164,165,166,167,168,169,170,171,172,173,174,175,176], when Dionigi et al. [177] presented a case of a 62-year-old Caucasian woman who had undergone five years of prior esophagectomy with en-bloc lymphadenectomy for a PMME. At subsequent checks, a chest x-ray revealed a 1.3 cm pulmonary nodule in the upper lobe of the left lung, which, when biopsied, was found to be a metastasis of a previous PMME. The authors, in this paper, stressed the concept of how important it was to perform a continuous follow-up over time for patients with previous PMME and, moreover, they remarked on what was reported by Chalkiadakis et al. [63], indicating a PMME metastasis rate of 78%.

After other reports [178,179,180,181,182,183,184,185,186,187], Fredricks et al. [188] reported an extremely rare case of PMME in a patient with human immunodeficiency virus (HIV) for more than 10 years. Interestingly, the operative esophagectomy specimen had the largest mass represented by the MM, but also two other pigmented lesions that were further away from the largest mass. On histological examination, these lesions resulted from junctional nevus and atypical melanocytic hyperplasia up to melanoma in situ. This case report has been of great importance because it has allowed us to provide a model of local tumorigenesis of PMME starting from dysplastic melanocytic lesions with greater accuracy.

In the following years, various authors continued to report an increasing number of cases of PMME [189,190,191,192,193,194,195,196,197,198,199,200,201,202,203,204,205,206,207,208,209,210,211,212,213,214,215,216,217,218] and, in 2012, Inadomi et al. described an interesting case of recurrent PMME in a 57-year-old patient who had already undergone subtotal esophagectomy for a PMME and who, five years later, in 2011, had recurrent melanoma at the point of previously conducted esophageal-jejunal anastomosis. This case report proved to be of great help in obtaining follow-up data, which in the case of patients with PMME were almost completely absent, considering that the 5-year survival rate is around 2.2%. Furthermore, another merit of this paper was to highlight the possibility that basic diagnostic instrumental methods, such as esophagoscopy, could not detect the entire extension of the PMME, as in the case of the patient in question, who presented melanocytic lesions atypical far beyond what was found in instrumental diagnostics [219]. These concepts were also analyzed in the next papers by Cardena [220] and Suarez-Aliaga et al. [221], and in a 2013 comparative study, Wang S. et al. presented the clinical data of 13 patients with PMME treated surgically as primary therapy at their center in the time period from 2000 to 2012. The mean age (± standard deviation) was 66.4 ± 7.6 years, and 84.6% were male. Overall, 61.5% of tumors were located in the lower thoracic esophagus. The authors reported surgical morbidity and mortality rates of 7.7% and 53.9%, respectively. The incidence of lymph node metastases for patients with tumors invading the mucosa was 0 but increased to 42.9% with tumors invading the submucosal layer. Furthermore, authors reported that PMME had a different probability to metastasize depending on the topography: for example, tumors localized in the middle third of the esophagus metastasized most frequently to the perigastric lymph nodes than the middle mediastinal lymph nodes. For PMME located in the lower third of the thoracic esophagus, superior mediastinal lymph node metastasis (2 of 4) with invasion of the tumor penetrating the correct muscle layer was more likely. Relapse occurred within 1 year in all patients with post-stage Ib cancer, and the most common recurrent organ was the liver. The 1- and 5-year overall postoperative survival rates were 54.0% and 35.9%, respectively, and lymph node metastases were the independent predictor for postoperative survival [222].

From 2013 to 2016, other 35 cases of PMME were reported in literature [223,224,225,226,227,228,229,230,231,232,233,234,235,236,237,238,239,240,241,242,243,244,245,246,247,248,249,250,251,252,253] with a particular emphasis about the case presented by Zhang et al. [228] that measured 12 cm (huge PMME).

In a 2016 paper, Gao S. et al. summarized and analyzed the characteristics of 17 patients with PMME (mean age of 57.5 ± 10.3 years) who had undergone surgical resection in their center. The majority of the patients were male, with smoking and alcohol consumption of 41.2% and 23.5%, respectively. The lymph node metastases in this series mainly involved the medium-lower mediastinal area and the upper abdominal area. PMMEs that invaded beyond the submucosal layer (T2–T4) had a much higher tendency to lymph node metastasis than those limited to the submucosal layer (T1) (6/8, 75.0% vs. 3/9, 33.3%, *p* = 0.086). The 1- and 5-year survival rates of the patients were 51% and 10%, respectively, with a median survival time of 18.1 months. Survival analysis showed that TNM stage was a predictor for PMME prognosis (median survival time of 47.3 months vs. 8.0 months for stage I/II versus stage III, respectively, *p* = 0.018) and multivariate Cox regression analysis revealed the independence of its prognostic value [HR (95% CI): 5678 (1125–28,658), *p* = 0.035] [254].

After other six cases of PMME [255,256,257,258,259,260], Hayian Su et al., in their 2017 work, analyzed 21 cases of PMME at Tianjin Medical University Cancer Institute and Hospital from January 2002 to February 2017. In terms of sintomatology, 19 patients presented with dysphagia and two patients complained of retrosternal pain. From a histological point of view, all tumors were composed of atypical melanocytes with a lot of melanin pigment intra and around the tumor. In this study, OS was 1–40 months, with the median time of 10 months. The mucosal staging classification for the upper aerodigestive tract showed a better distribution of overall survival with different stages than that of the American Joint Commission on Cancer staging classification for esophagus, but without statistical difference. Both the clinical and pathological characteristics were not highly consistent with OS [261].

In this way, after the some other papers [262,263,264], it is important to mention a contribution from Katerji et al. who, again in the same year, presented a case of PMME in an 82-year-old man in which the immunophenotype was aberrant for the CD56 (or NCAM, Neural Cell Adhesion Molecule) marker. The authors, in fact, warned against making a hasty diagnosis as it is the pathologist’s work to know these potentially dangerous pitfalls [265].

It is important to underline that among the PMMEs, a rate of between 15 and 25% is present as amelanotic lesions [83,106,171,172,174,178,194,266,267,268,269,270,271,272,273,274,275]. Kobayashi et al., among other authors, reported a case of amelanotic malignant melanoma of the esophagus in a 68-year-old man previously operated on for the detection of gastric carcinoma. After a few months, following the finding of dysphagia symptoms, the patient underwent esophagoscopy, which revealed a mucosal melanosis adjacent to a protruding lesion in the esophageal lumen. The patient underwent a curative trans-thoraco-abdominal subtotal esophagectomy. Immunohistochemical examination of the excised sample was negative for markers such as HBM-45 and Melan-A but was positive for SOX10 (Sry-related HMg-Box gene 10), KBA.62 and tyrosinase. A primary diagnosis was made of an amelanotic malignant melanoma of the esophagus which consisted only of premelanosomes. The authors’ findings suggest that, in the diagnosis of malignant melanoma, SOX10 and KBA.62 may be useful, particularly in the diagnosis of amelanotic malignant melanoma [271].

Regarding one of the largest cohorts of PMME patients, Wang et al. conducted a retrospective analysis of clinical data from 76 patients at Peking University Cancer Hospital over the period January 2008 to September 2017. The authors assessed objective response rates (ORR) and progression-free survival (PFS). These patients were classified as unresectable or metastatic and were assigned to three cohorts based on the treatment they underwent: chemotherapy (C: 46 patients), targeted therapy (T: 2 patients) and PD-1 inhibitors (IT: 12 patients). In this study, PFS in cohort C was three months with a limited ORR of 10.9%. In the IT cohort, seven patients (75.0%) achieved a partial response and three had stable disease for more than four months. The median PFS in the IT cohort was not reached and the mean was 15.6 months, which was much longer than in cohort C (*p* < 0.001) [276]. Therefore, these authors renewed the need to employ techniques such as massive gene sequencing to be able to search for biomarkers capable of predicting the response of PMME to anti-PD1 immunocheckpoint inhibitors. After another 38 cases of PMME [274,277,278,279,280,281,282,283,284,285,286,287,288,289], Endo F. et al. [290] reported a case of a 70-year-old man who complained of dysphagia due to the presence of a polypoid lesion in the lower thoracic esophagus. Although the histopathological examination had provided the diagnosis of esophageal squamous carcinoma at first, with subsequent pT3N1M0 staging and clinical stage III, a histological re-evaluation after radical esophagectomy revealed the presence of atypical melanocyte cells positive for S-100 protein, Melan-A and HMB-45, and, therefore, PMME was diagnosed. After 16 months, an abdominal TC revealed a solitary retroperitoneal recurrence in the lateral layer of the ascending colon. FDG-PET imaging showed hypermetabolic accumulation with a SUV value of 5.8. The patient was treated with nivolumab (240 mg) every two weeks and after eight courses of nivolumab, the abnormal accumulation of retroperitoneal mass disappeared on PET scan and this therapeutic effect continued for 20 months.

Data very similar to this was reported by Ito S. et al. [291], who presented a case of an 81-year-old Japanese female presented with a PMME diagnosis histologically established. After the surgery, the patient presented multiple lymph node and bone metastases, and so began treatment with nivolumab 240 mg every two weeks. After eight cycles of nivolumab, both lymph node and bone metastases were markedly reduced. The patient received 30 courses of nivolumab and has maintained the partial response without severe adverse events. These data are very similar to those of the paper of Chu et al. [292].

Very interesting, again in this context, is the case reported by Yamaguchi T. et al. [293], in which the authors describe the so-called “abscopal” effect in a 75-year-old woman with a PMEE of the esophagogastric junction, removed by proximal gastrectomy with resection of the lower esophagus. After three months, the patient complained of left pleural metastases and began nivolumab, but after three courses, the patient exhibited grade 3 renal dysfunction, as a result of which the drug was discontinued. Five months after the development of renal dysfunction, a CT scan showed a non-colored lump within the pancreas and the patient was diagnosed with pancreatic metastasis; intensity modulated radiotherapy was performed. Six months later, CT revealed that the pancreatic nodule and pleural metastases had decreased; after another two months, the pleural metastases and the effusion had disappeared. The patient was alive and in clinical remission.

This paper has the merit to describe a concomitant “synergistic” effect between irradiation (radiotherapy) and the administration of nivolumab, setting a rationale for future studies capable of studying the potential use of both therapeutic modalities.

Surgery has traditionally been the only option for prolonging survival in patients with PMME [146,169,213,235,247,259,294], and total or sub-total esophagectomy appears to offer better survival outcomes. The efficacy of both chemotherapy and radiotherapy have been shown to be limited [166,167,201,218,241,260]. The introduction of immunotherapy, particularly programmed death 1 (PD-1) inhibitors such as nivolumab and pembrolizumab, together with the cytotoxic T-cell-associated antigen 4 inhibitor (CTLA4) such as ipilimumab in combination with nivolumab, significantly extended overall survival in metastatic melanoma. Indeed, ICI is currently positioned as a first-line therapy for melanoma without BRAF mutations, and the 3-year overall survival rates have reached 51%, 40% and 56%, respectively [295].

Although Nivolumab has been shown to be less effective in mucosal melanomas than in cutaneous melanomas, the standard treatment and efficacy of ICI in PMME remain unclear due to its rarity [275,288,289,290,291,292,293,296,297,298,299,300,301,302,303,304]. As we have analyzed, some studies have revealed promising results, but there is a need for randomized controlled trials with larger series of PMME patients [186,218,275,305].

Table 1 summarizes PFS and OS in the studies published so far regarding the use of ICI in PMME.

In recent years, the greater diffusion of molecular biology techniques such as next generation sequence (NGS) and parallel massive sequencing have made it possible to deepen the knowledge related to PMME by analyzing the different mutations responsible for this entity [275,289,291,292,293,296,297,298,299,300,301,302,303]. More specifically, in a very recent work, Tsuyama et al. studied 13 cases of PMME and 10 cases of malignant skin melanoma (SKMM) with NGS and immunohistochemistry. NGS analysis based on DNA and RNA hybridization capture revealed that NF1 was the most frequently mutated gene, but other mutations detected were SF3B1, KRAS, BRCA2, KIT and TP53.

Furthermore, the authors indicated that the common BRAF mutation (present in cutaneous malignant melanoma) was not present in the analyzed cases of PMME. It was also agreed that a subset of PMME may contain viable mutations. Furthermore, the authors reported that response rates to immunotherapy were lower in this patient cohort [304]. Finally, immunohistochemistry results and mutation status were concordant between p53/c-Kit and TP53/KIT, with a focal expression of PD1 observed in a single PMME sample. The burden of tumor mutations in PMME was significantly lower than that in SKMM (*p* = 0.030), and any PMME cases showed high microsatellite instability. RNA sequencing revealed a distinctive pattern with respect to RNA expression. T cell co-stimulation differed between PMME and SKMM. These data show that the RAS-mitogen-activated protein kinase pathway is one of the main pathways involved in PMME.

## 5. Conclusions

PMME represents a very rare entity whose very existence has been the subject of debate for a long time. Over time, an increasing number of cases have been reported in the literature, leading to an increase in knowledge and laying the foundations for a discussion on the treatment of this pathology, which still remains largely represented by surgery. In recent times, the possibility of discovering greater mutations in gene hotspots has made it possible to develop new therapeutic strategies, of which nivolumab is an example. Future studies with large case series, with clinicopathological and molecular data, will be necessary to improve the outcome of patients with PMME.

## Figures and Tables

**Figure 1 cancers-14-03725-f001:**
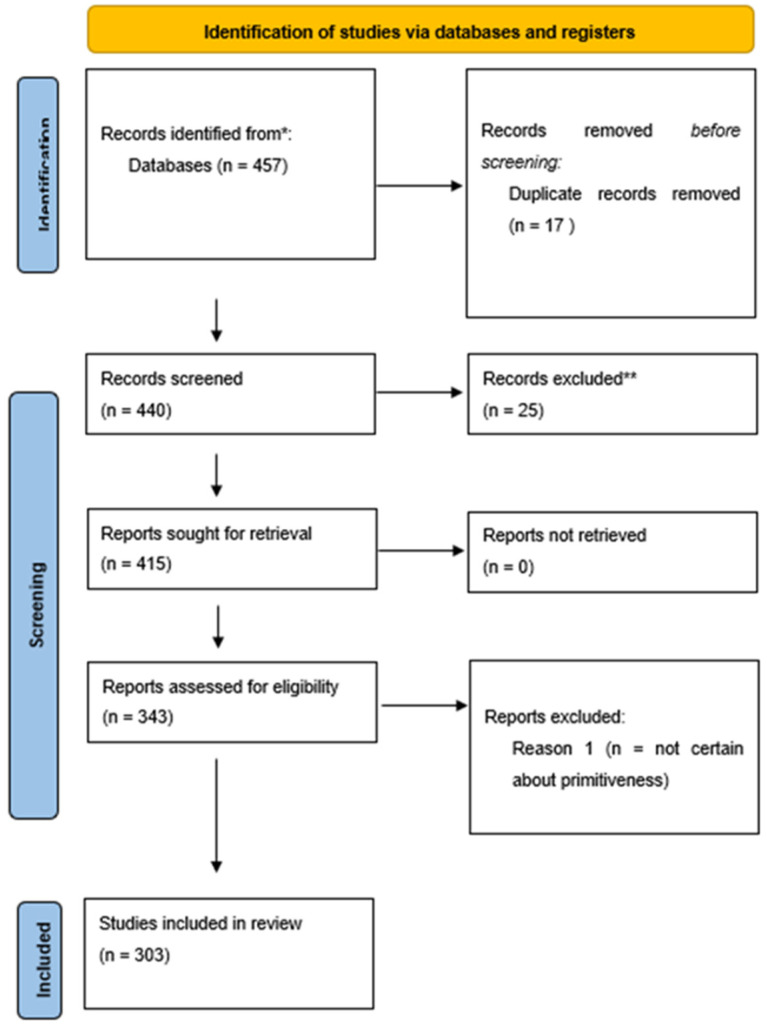
PRISMA 2020 flow-chart utilized in this review.

**Table 1 cancers-14-03725-t001:** Overall Survival and Regression-free Survival data from some studies with a great number of patients with PPME.

Author(s)	Number of Patient	Treatment	Overall Survival (OS)	Recurrence-Free Survival (RFS)	Future Prospectives
Hashimoto [306]	6	Surgery 4 ICI (2)	19.6 (6.4–40.5) months	19.3 months (range, 3.9–37.9)	Additional studies
Wang [276]	76	Surgery 59 Adjuvant therapy 37	22.3 months	4.5 months	PD-1 valuable option for therapy
Lasota [278]	16 (data available)	Surgery 16	4–22 months	/	New studies about NGS
Cheng [281]	9	Surgery 5 PR 1 Chp 1 ICI 1 Biotherapy 1	OS stage I: 100% OS stage OS stage III-IV: 0%	/	More randomized controlled trials
Dai [285]	70	Surgery/adjuvant chemotherapy	13.5 months	/	/
Kim [302]	17	Surgery 10 Chemotherapy/ICI 7	10 months	4 months	Further large-scale studies are required regarding novel treatment strategies such as immunotherapy for patients with PMME

Legend. PR: Palliative Resection; Chp: chemotherapy post-operatory; ICI: Immunocheckpoint inhibitors.

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
