# Peer review of "The Thousand Faces of Malignant Melanoma: A Systematic Review of the Primary Malignant Melanoma of the Esophagus"

_cancers, 2022, doi:10.3390/cancers14153725_

Round 1
Reviewer 1 Report
Thankyou to the authors for addressing the corrections so quickly. Unfortunately there are still major issues with the references with many missing and others reported out of order. I suggest the authors print the reference list and cross off the list each time a reference appears in the manuscript. As the manuscript states that 303 publications were included in the study the authors will then need to read all missing papers and add reference to them where appropriate.
I apreciate that the authors made every attempt to have reference 3 accessible however I still cannot access it.
Author Response
Dear Reviewer n'1, thank you very much for your suggestions aimed, once again, at improving the quality of our paper. We printed the work and followed his advice: reference after reference we checked that they were all correctly cited in order and that none were missing. We added phrases for those references that we had not properly cited but which, over the months, we had read ALL. Furthermore, we have decided to replace the reference number 3 with a more current one, in order to make it easier for you to check it. We hope that now the work can go well. Best wishes and thanks again,
Reviewer 2 Report
The authors have addressed previous comments and added additional information that solidifies the manuscript
Author Response
Dear Reviewer n'1,
thank you very much!
The authors
Reviewer 3 Report
Given that the concerns raised by the reviewers have been properly addressed, the manuscript is recommended to be published as it is.
Author Response
Dearest Reviewer n'3,
Thank you very much!
The authors
Round 2
Reviewer 1 Report
Thank you authors for adding in the missing references.
This manuscript is a resubmission of an earlier submission. The following is a list of the peer review reports and author responses from that submission.
Round 1
Reviewer 1 Report
The manuscript is mostly well written however it is lacking in discussion of the papers. For example, on page 6, there is a summary of a paper presenting treatment outcome for patients on PD-1 inhibitors. The authors then list 4 more anti-PD-1 therapy papers. Why not discuss what the average OS or PFS is for all 5 studies. Do the 5 together indicate that anti-PD-1 is effective or ineffective for treating PMME?
It is stated in the introduction that the purpose of the review is “focusing on the new therapeutic strategies …trying to designate future perspectives on the treatment of this entity”. However, very little attention is paid to treatment. Perhaps a summary table of treatments and response or survival rates would provide a point of discussion, rather than just listing treatments that have been tried as currently in the final paragraph of the discussion.
There are 305 references in the bibliography but only 186 included in text. Some in text citations are also out of order.
I was unable to find reference 3 and a question if it provides both an estimate of PMME incidence and guidelines for evidence based medicine.
Author contribution is incomplete.
Typographical errors:
Last paragraph of pg 3 and first of pg 4 are repeated.
Within the repeated section there is an extra ‘.’.
Pg 4, 168. Typo. Bu should be by.
Pg 5, 192. Exclude should be excluding.
Pg 5, 193 extra ‘.’
Pg 5, 198. Incomplete sentence.
Pg 5, 200. Repeated word, had.
Pg 5, 219. In situ should be in italic.
Pg 7, 305. ‘.in PPME.’ Needs to be deleted.
Pg 7, 311. Incomplete sentence.
In text references have names in italic, for consistency the following instances should be corrected. However, I not this is not consistent with the style used in Cancers and suggest that instead italics should be removed from all other references.
*Pg 5, 196. Waller should be in italic.
*Pg 5, 214. Fredricks should be italic.
*Pg 5, 222. Indaomi should be in italic.
Reviewer 2 Report
This is a well written review manuscript titled "The thousand faces of malignant melanoma: a comprehensive review of the Primary Malignant Melanoma of the Esophagus."
The recommendation I have for the authors is to submit a new table with the results section in terms of the % of patients and the N number of each result section.
Reading through %s in such a study is helpful, but it will be more helpful to see how many times that has been documented in the reviewed studies along with the %. the table format will make the results easier to read.
Reviewer 3 Report
Manuscript by Cazzato et. al. provides comprehensive review of rare primary malignant melanoma of esophagus (PMME). In this review they provide combine study of 303 article publish in various journals which include 347 patients with PMME.
Author provides some details about various therapeutics strategies (like pharmacological drug and possible immunotherapy). As new study author tries to provides details study about PMME, however various other publish article (review) provide much details about PMME and treatments.
This manuscript lacks novelty and future prospective about PMME.